# Whole genome sequence analysis reveals the broad distribution of the RtxA type 1 secretion system and four novel putative type 1 secretion systems throughout the *Legionella* genus

Connor L. Brown[1,2], Emily Garner[1,3], Guillaume Jospin[4], David A. Coil[4], David O. Schwake[5], Jonathan A. Eisen[4,6], Biswarup Mukhopadhyay[2], Amy J. Pruden[1]*

**1** Via Department of Civil and Environmental Engineering, Virginia Tech, Blacksburg, VA, United States of America, **2** Department of Biochemistry, Virginia Tech, Blacksburg, VA, United States of America, **3** Department of Civil and Environmental Engineering, West Virginia University, Morgantown, WV, United States of America, **4** Genome Center, University of California, Davis, CA, United States of America, **5** Department of Natural Sciences, Middle Georgia State University, Macon, GA, United States of America, **6** Evolution and Ecology, Medical Microbiology and Immunology, University of California, Davis, CA, United States of America

* apruden@vt.edu

**Data Availability Statement:** All data including all Legionella sequences identified in the present

## Abstract

Type 1 secretion systems (T1SSs) are broadly distributed among bacteria and translocate effectors with diverse function across the bacterial cell membrane. *Legionella pneumophila*, the species most commonly associated with Legionellosis, encodes a T1SS at the *lssXY-ZABD* locus which is responsible for the secretion of the virulence factor RtxA. Many investigations have failed to detect *lssD*, the gene encoding the membrane fusion protein of the RtxA T1SS, in non-*pneumophila Legionella*, which has led to the assumption that this system is a virulence factor exclusively possessed by *L. pneumophila*. Here we discovered RtxA and its associated T1SS in a novel *Legionella taurinensis* strain, leading us to question whether this system may be more widespread than previously thought. Through a bioinformatic analysis of publicly available data, we classified and determined the distribution of four T1SSs including the RtxA T1SS and four novel T1SSs among diverse *Legionella* spp. The ABC transporter of the novel *Legionella* T1SS *Legionella* repeat protein secretion system shares structural similarity to those of diverse T1SS families, including the alkaline protease T1SS in *Pseudomonas aeruginosa*. The *Legionella* bacteriocin (1–3) secretion systems T1SSs are novel putative bacteriocin transporting T1SSs as their ABC transporters include C-39 peptidase domains in their N-terminal regions, with LB2SS and LB3SS likely constituting a nitrile hydratase leader peptide transport T1SSs. The LB1SS is more closely related to the colicin V T1SS in *Escherichia coli*. Of 45 *Legionella* spp. whole genomes examined, 19 (42%) were determined to possess *lssB* and *lssD* homologs. Of these 19, only 7 (37%) are known pathogens. There was no difference in the proportions of disease associated and non-disease associated species that possessed the RtxA T1SS (p = 0.4), contrary to the current consensus regarding the RtxA T1SS. These results draw into

study are provided in the paper and its supplementary materials, and may be accessed online through FigShare at doi:10.6084/m9. figshare.9162161.v1, doi:10.6084/m9.figshare. 9162146.v1, doi:10.6084/m9.figshare.9162152.v1, doi:10.6084/m9.figshare.9162161.v1. The Whole Genome Shotgun project has been deposited at DDBJ/ENA/GenBank under the accession PRJNA450138. The version described in this paper is version PRJNA450138.

**Funding:** This study was supported by the US National Science Foundation via RAPID Award no. 1556258: Recipient: A.P.; a Graduate Research Fellowship Program Grant (DGE 0822220): Recipient: E.G.; Supplementary funding associated with award #1336650: Recipient: A.P. Additional support was provided by the Alfred P. Sloan Foundation Microbiology of the Built Environment program (https://sloan.org/programs/research/microbiology-of-the-built-environment#apply): Recipient: E.G.; the American Water Works Association Abel Wolman Fellowship: Recipient: E. G.; and the Institute for Critical Technology and Applied Science and Engineering of the Exposome Center (https://ictas.vt.edu/) at Virginia Polytechnic Institute and State University: Recipient: A.P. In addition to the funding sources noted, colleague Marc A. Edwards contributed discretionary research funding to support the sequencing study. The funders had no role in study design, data collection and analysis, decision to publish, or preparation of the manuscript.

**Competing interests:** The authors have declared that no competing interests exist.

question the nature of RtxA and its T1SS as a singular virulence factor. Future studies should investigate mechanistic explanations for the association of RtxA with virulence.

## Introduction

Type 1 secretion systems (T1SSs) are broadly distributed among bacteria and mediate the translocation of protein or peptide substrates with a broad range of function [1–4]. The core T1SS complex is composed of a dimerized inner membrane ATP-binding-cassette (ABC) transporter protein, a trimerized membrane fusion protein that spans the periplasm, and a trimerized outer membrane protein. The genes encoding the ABC transporter and membrane fusion protein are typically localized together in the genome [5], while the gene encoding the outer membrane protein is usually encoded elsewhere, reflecting the multifunctional nature of this family of proteins [6,7].

Three major classes of T1SSs can be described based on the N-terminal region of the ABC transporter [4,8] (Fig 1). The first class includes bacteriocin transporters, such as the colicin V system in *Escherichia coli* [10], which encode ABC transporter proteins with N-terminal C-39 peptidase domains that cleave N-terminal regions of nascent substrates during translocation [4] (Fig 1A). In another class, such as the HlyA secretion system in *E. coli* [11], the ABC transporters possess an N-terminal C-39 peptidase-like domain (CLD), which lacks the catalytic histidine [10] (Fig 1B). A third class of T1SSs are composed of ABC transporters that lack either the C-39 peptidase or CLD. These systems typically secrete smaller substrates, including epimerases and proteases in *Azotobacter vinelandi* and *Pseudomonas aeruginosa*, respectively [12,13] (Fig 1C).

In *Legionella pneumophila*, the prototypical T1SS is encoded at the *lssXYZABD* locus with *lssB* and *lssD* encoding the ABC transporter and membrane fusion protein, respectively [14,15]. This complex is responsible for secreting the virulence factor RtxA, which is associated with adherence, pore-formation, cytotoxicity, and entrance into host cells [16,17]. The RtxA T1SS belongs to a subset of the CLD-type T1SSs whose substrates possess an N-terminal retention module. The most well characterized system of this type is the adhesin LapA in *Pseudomonas fluorescens* strain Pf01 [3,18]. In *P. fluorescens*, LapA is secreted in a two-step fashion mediated by membrane fusion protein LapC, the ABC transporter LapB, outer membrane protein LapE, and a transglutaminase-like cysteine proteinase (BTLCP) LapG [3,18–20] (Fig 1D). However, no study has detected *lssD* (a *lapE* homolog) in any non-*pneumophila* genome since the discovery of the *lssXYZABD* locus in 2001 [15,17,21]. These observations have led to the assumption that RtxA is a key virulence determinant unique to *L. pneumophila*. In the present study, we report the broad distribution of the RtxA T1SS and four putative novel T1SSs throughout the *Legionella* genus and examine their occurrence among strains of disease-associated *Legionella*.

## Results

### Isolation and phylogenetic identification of a novel *Legionella taurinensis* strain containing four type 1 secretion systems

During a survey of municipal and well waters in Genesee County, Michigan, endemic *L. pneumophila* were targeted for isolation using standard culture methods, and isolates were subjected to whole genome sequencing [22]. Sequence and phylogenetic analysis revealed four

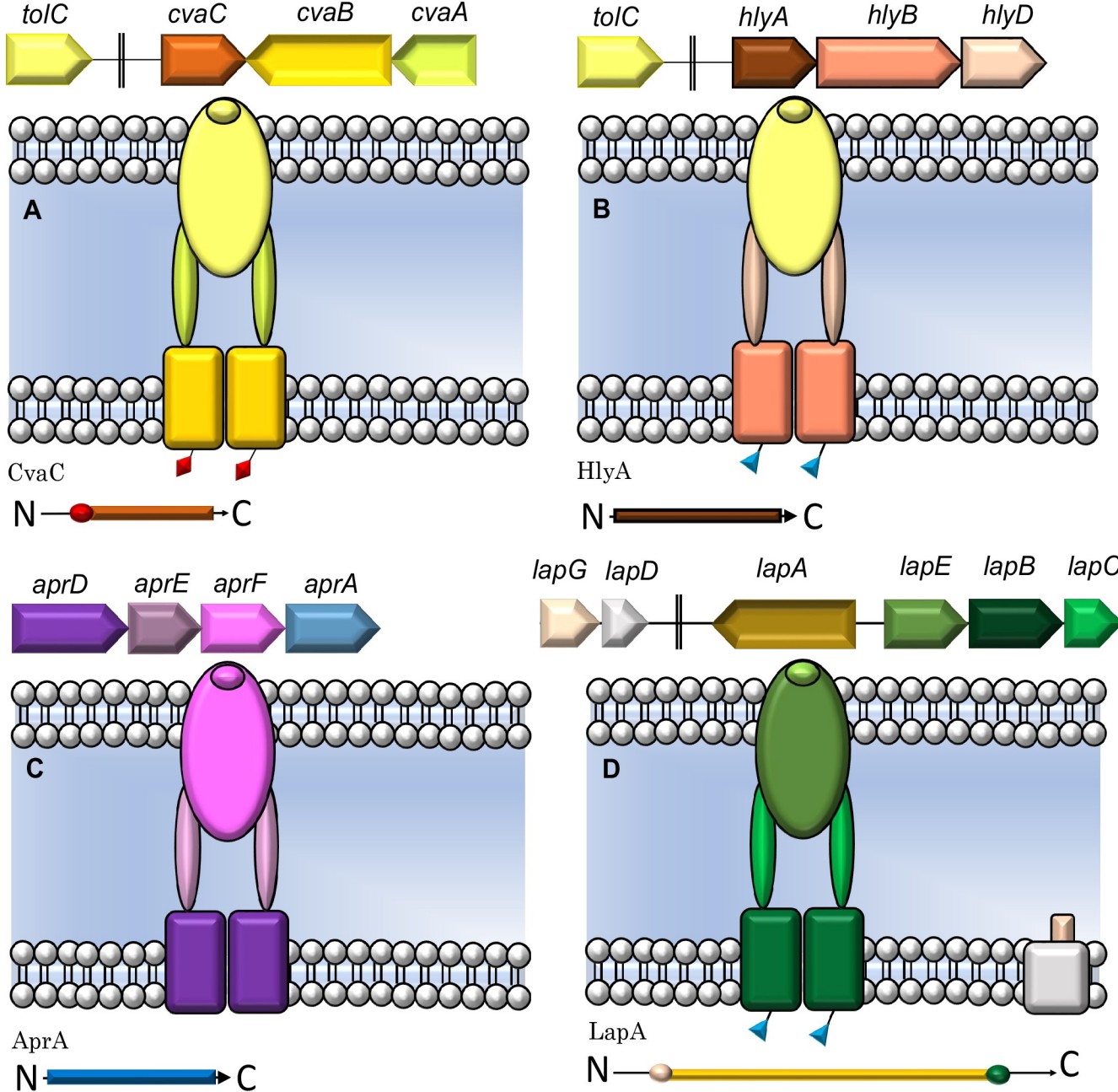

**Fig 1. Genomic organization and structure of three type 1 secretion systems with corresponding substrates below each system [3, 8, 9].** (A) The CvaC T1SS in *E. coli*. An N-terminal region of CvaC is cleaved during secretion by the C-39 peptidase motifs on CvaB (red diamonds). (B) The HlyA T1SS in *E. coli*. HlyB has the CLD domain (blue triangles). (C) The AprA T1SS. AprD lacks either the CLD or C-39 peptidase motif. (D) The LapA T1SS in *P. fluorescens* Pf01. LapA is secreted in a two-step fashion where the retention module (tan circle on the N-terminus of LapA) is cleaved by LapG before release into the extracellular environment.

isolates obtained from municipal water sourced from an aquifer to be novel *Legionella taurinensis* strains [23], a species that did not have a reference genome at that time (S1 File Table A). These genome sequences are deposited at DDBJ/ENA/GenBank under the accession PRJNA450138. While analyzing the genomes of the isolates for virulence factors, the strain was found to possess the *lssXYZABD* locus believed to be absent from non-*pneumophila*

*Legionella* spp. [15,17,21,24] in addition to three novel T1SSs with low homology to the RtxA T1SS components.

### Organization of the *lssBD* system in *Legionella* spp.

In contrast to previous reports [15,21], *L. taurinensis* was found to possess the entire *lssXY-ZABD* locus associated with RtxA secretion with two reading frames (ORFs) (472 base pair and 4.24 kilobase) between *lssB* and *lssA*, both of which encode hypothetical proteins (Fig 2A and 2B and S1 File Figs A and B). *L. taurinensis* and *L. pneumophila* LssB possess the LapB-type N-terminal CLD (S1 File Fig B, S1 File Table B). *L. taurinensis* possesses the *lssXYZABD* locus including a gene encoding an LssD homolog which is 60% identical to *L. pneumophila* LssD (S1 File Fig A).

Following this observation, we examined 45 *Legionella* species whole genome sequences for the RtxA T1SS by comparing amino acid sequences of *L. pneumophila* LssB and LssD against the predicted proteomes of *Legionella* spp. using blastp (S1 File Table B). A species was considered to encode the RtxA T1SS if its genome encoded homologs of LssD and LssB with amino acid sequence $\geq$ 40% identity (S1 File Table B) and if the ABC transporter was monophyletic with *L. pneumophila* LssB (Fig 3, S1 File Figure C). These two proteins were chosen as they constitute two-thirds of the core components of a T1SS, the membrane fusion protein and ABC transporter. This analysis revealed that nearly half of species' genomes examined (n = 19) encoded LssB and LssD homologs. Of these 19, several possessed the entire *lssXYZABD* locus, while others lacked some components of the locus or possessed additional ORFs within the cluster. In all in which the RtxA T1SS was detected, *lssB* and *lssD* were encoded adjacent to one another in the genome.

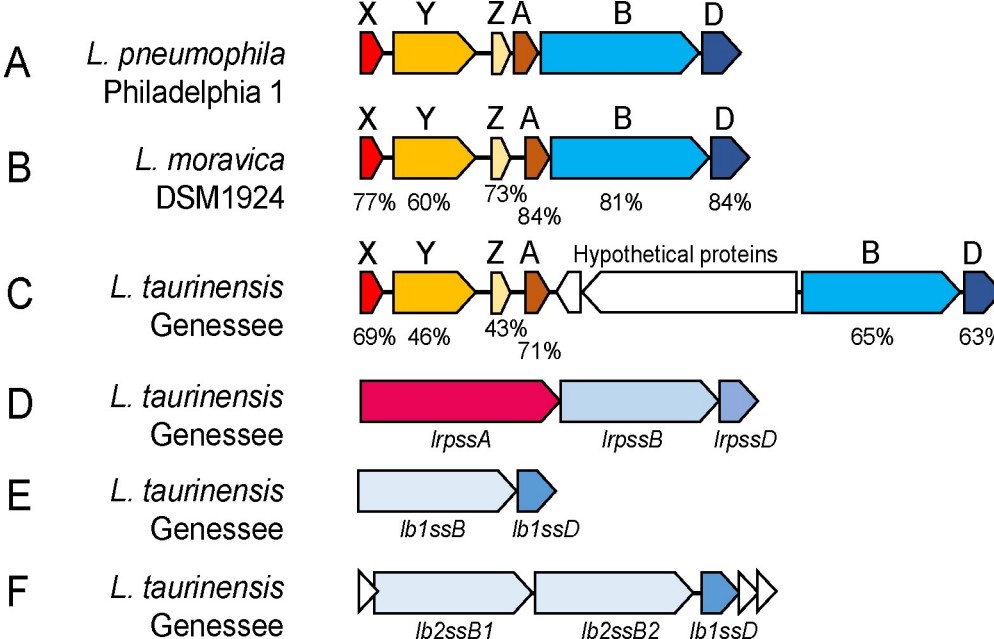

**Fig 2. Genomic organization of four type 1 secretion systems in the *Legionella* genus.** (A) Organization of the *lssXYZABD* locus in *L. pneumophila*. (B) The *lssXYZABD* locus in *L. moravica* DSM1924.(C) The *lssXYZABD* locus in *L. taurinensis* encodes homologs of all members of the *lssXYZABD* locus, but the operon has two inserted ORFs. The *lssXYZABD* locus in *L. moravica* DSM1924 (D) The *lrpss* T1SS found in *L. taurinensis* encodes a putative substrate at the LRPSS locus. (E and F) The *lb1ss* and *lb2ss* locus in *L. taurinensis*. White triangles are hypothetical proteins. (B-C) Percentages are percentage identity to *L. pneumophila* Philadelphia amino acid sequences. GenBank accession numbers are provided (S1 File Table B).

*Legionella moravica* strain DSM19234 was found to possess the entire *lssXYZABD* locus with corresponding proteins 60–86% identical to those encoded by *L. pneumophila* Philadelphia 1 (Fig 2A and 2C, Table 1). This is noteworthy as a previous bioinformatic investigation reported the absence of this locus in *L. moravica* DSM19234 in its entirety [24]. This study reports using the tblastn algorithm (release 2.2.25) to compare the protein sequences of the locus with nucleotide sequences of *Legionella* spp. Repeating this approach using tblastn, *L. moravica* DSM19234 whole genome sequences (taxid: 1122165), were found to possess all genes of the *lssXYZABD* locus with identity values ranging from 56.28%-84.13% (Table 1). Therefore, whole genome sequences of *L. moravica* DSM19234 were found to possess genes encoding the RtxA T1SS. Last, we additionally confirmed the presence of an RtxA-like substrate in a subset of genomes, including those of *L. taurinensis* Genessee01 and *L. moravica* species, by identifying T1SS substrates with a putative N-terminal di-alanine retention module specific to LapA/RtxA class substrates (S1 File Figure D) [3,18] through tblastn searches against their whole genome sequences.

*Legionella longbeachae* is the second most commonly reported causative agent of Legionnaires' Disease, especially in New Zealand and Japan where reported cases of *L. longbeachae* infection occur about as often as cases of *L. pneumophila* infection [25]. Draft genome sequences of *L. longbeachae* strains F1157CHC and FDAARGOS-201 include *lssB* and *lssD* homologs with interrupting stop codons within the ORFs (S1 File Table B) indicating that these genes are non-functional or contain sequencing or annotation errors. A tblastn search with LssB and LssD from *L. longbeachae* strain FDAARGOS-201 as queries did not identify respective homologs in the genome of *L. longbeachae* strain NSW150 (E-value cut-off of 1E-5). Therefore, the only finished genome of *L. longbeachae*, strain NSW150, lacks homologs of both *lssB* and *lssD*. Further, all three genomes of *L. longbeachae* were examined for homologs of the *L. pneumophila* Philadelphia 1 RtxA (lpg0645) and LapE (lpg00827) using tblastn and neither were detected (E-value cut-off 1E-5).

## *L. taurinensis* encodes three novel type 1 secretion systems

Analysis of the diversity of T1SSs across the *Legionella* genus has not been previously reported. Scanning the genome of *L. taurinensis* Genessee01 for additional *lssB* and *lssD* family genes revealed the presence of three putative novel T1SSs. One is composed of a 438 amino acid (aa) membrane fusion protein and 587 aa ABC transporter which were encoded adjacent to one another in the genome (Fig 2D). This ABC transporter lacks either the C-39 peptidase motif or CLD (S1 File Figs E and F) and a protein phylogeny of the ABC transporter indicated a relationship with T1SSs that secrete substrates of diverse function (Fig 3, S1 File Fig C). Additionally, the genomic locus in *L. taurinensis* that encodes the ABC transporter and membrane fusion protein also encodes a putative substrate with hemolysin type calcium binding motifs commonly associated with T1SS substrates. Because of this, the name *Legionella repeat protein secretion system* (LRPSS) is proposed for this T1SS.

*L. taurinensis* additionally encodes two putative bacteriocin transport T1SSs. One is a colicin V-like T1SS (Fig 2F, Fig 3), for which the name *Legionella bacteriocin 1 secretion system* (LB1SS) is proposed. The second putative bacteriocin transporter locus encodes two ABC transporter proteins at the same genomic locus, (Fig 2E and Fig 3, S1 File Fig E and S1 File Table B), with one of these (PUT41641.1) including a C-39 peptidase motif at the N-terminus. This ABC transporter was found to be phylogenetically related to a nitrile hydratase leader peptide (NHLP)-type bacteriocin ABC transporter in *Nostoc sp. PCC 7120* [26] (Fig 3, S1 File Figs C and E). For this system, the name *Legionella bacteriocin 2 secretion system* (LB2SS) is proposed. Additionally, searching the genomes of the *Legionella* spp. for homologs of the

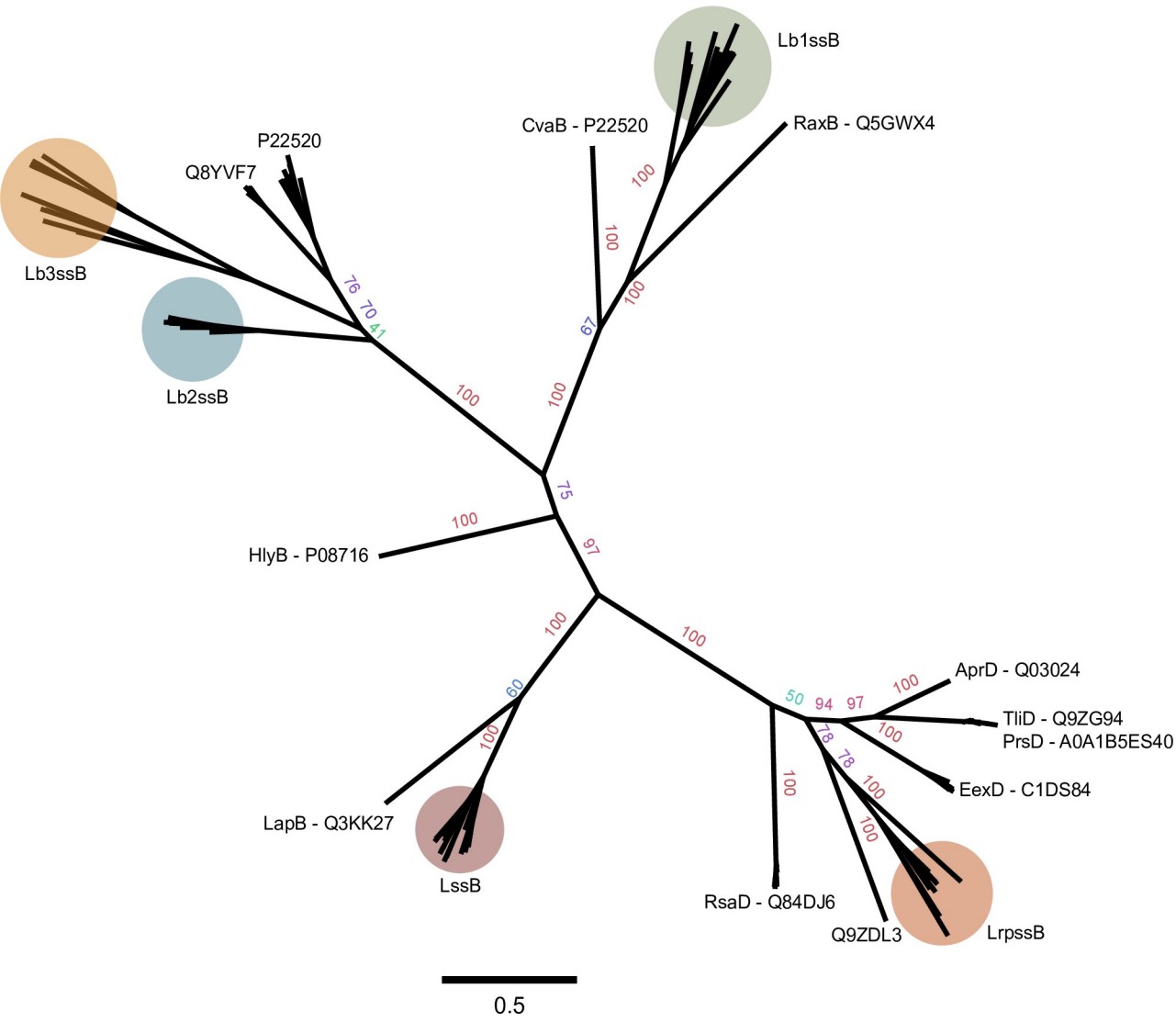

**Fig 3. Unrooted maximum likelihood tree for trimmed sequences of 187 T1SS ABC transporters including the five *Legionella* systems.** Bootstrap values displayed as percentages. An un-modified tree used to classify the *Legionella* secretion systems, and the *Legionella* sequences, are provided (S1 File Fig E and Table B). Pink boxes indicate an association with disease.

LB2SS system genes revealed a similar NHLP-type bacteriocin T1SS. This was suggested by protein phylogeny (Fig 3, S1 File Fig C) to be a different T1SS or a highly diverged variant of

**Table 1. *L. moravica* DSM19234 posesses the RtxA T1SS. Components were detected by tblastn. Identity values are adjusted for query cover.**

| Query Protein | Accession ID | Subject | tblastn Identity |
|---|---|---|---|
| LssB | CAD90959.1 | L. moravica DSM19234 | 76.10% |
| LssD | CAD90958.1 | L. moravica DSM19234 | 84.13% |
| LssA | CAD90960.1 | L. moravica DSM19234 | 83.95% |
| LssZ | CAD90961.1 | L. moravica DSM19234 | 66.50% |
| LssY | CAD90962.1 | L. moravica DSM19234 | 56.28% |
| LssX | CAD90963.1 | L. moravica DSM19234 | 77.27% |

the LB2SS. The name *Legionella bacteriocin 3 secretion system* (LB3SS) is proposed for this system.

## Distribution of T1SSs in the *Legionella* genus and their association with disease

We found that 19 of 45 (42%) *Legionella* spp. examined encode the RtxA T1SS (Fig 4). Of these 19, 7 (37%) are known pathogens [27]. Relatively few (10 of 45, 22%) encode the LRPSS system, while 16 of the 45 species (35%) encode the LB1SS; 9 of 45 species (20%) encode the LB2SS, and 11 of 45 species (24%) encode the LB3SS. Collectively, only eight species lacked any of the T1SSs described in this paper. We performed significance testing because previous studies have qualitatively evaluated differences in the distribution of T1SSs in relation to perceived virulence or propensity for intracellular growth. Two proportion Z-tests with continuity correction were performed to compare the prevalence of the RtxA T1SS within strains who have never been isolated from patients and those which have (Fig 4). No significant differences were detected (p = 0.4), but the sample sizes are small (n = 19). Thus, there was no detected difference in the proportions of disease and non-disease associated species that possessed any of the T1SSs to the extent of our current knowledge of pathogenicity in these *Legionella* species. However, our observation brings up the possibility that some of the *Legionella* species that thus far have not been isolated from patients are inefficient colonizers of built environments that are common exposure routes and, therefore, remain classified as non-pathogenic. Additionally, increasing the number of genomes analyzed could impact the results of the present analysis.

## Discussion

Multiple investigations report the restriction of the *lssBD*/RtxA system to *L. pneumophila*. Two studies did not detect *lssD* in non-*pneumophila Legionella* using Southern blotting and DNA macroarrays, respectively [15,21], while one study detected *lssB* in all non-*pneumophila Legionella* tested (n = 10) [15]. Incidentally, the presence of *rtxA* (determined by Southern blotting) in *Legionella feeleii* has been reported, but the study did not test for the presence of the T1SS components [17]. In retrospect, it may be unsurprising that several early studies did not detect the presence of *lssD* in non-*pneumophila Legionella* spp. given their reliance on DNA-DNA hybridization methods [15,16,21]. This component may be more variable due to the nature of its interactions with a rapidly evolving substrate, and therefore methods relying on the nucleotide sequence of *L. pneumophila lssD* as probe could plausibly cause false-negatives of this nature. On the other hand, *Qin et al* 2017 bioinformatically examined non-*pneumophila Legionella* genomes (n = 21) for the *lssXYZABD* components [24] and reported the absence of the *lssXYZABD* locus from all strains tested, including *L. moravica* strain DSM19234. Further, this study reported that *Legionella* lacking the *lssXYZABD* locus displayed reduced intracellular multiplication relative to *L. pneumophila* strains which possess the T1SS [24]. Thus, the emergent consensus model has been that the RtxA system is an important conserved genetic virulence determinant unique to *L. pneumophila*, despite the absence of additional experimental investigations of its function in animal models since the discovery of the enhanced entry locus in 2001 [15]. In contrast, we found bioinformatic evidence that homologs of the RtxA T1SS and the four novel T1SSs are prevalent throughout the genus, even among species not presently known to cause disease.

The 2014–2015 Center for Disease Control (CDC) Legionnaires' Disease Surveillance Summary Report [28] documents the species isolated in culture confirmed Legionnaires' Disease cases in the United States. Of 307 culture-confirmed cases in 2014–2015, 186 (60.6%) were

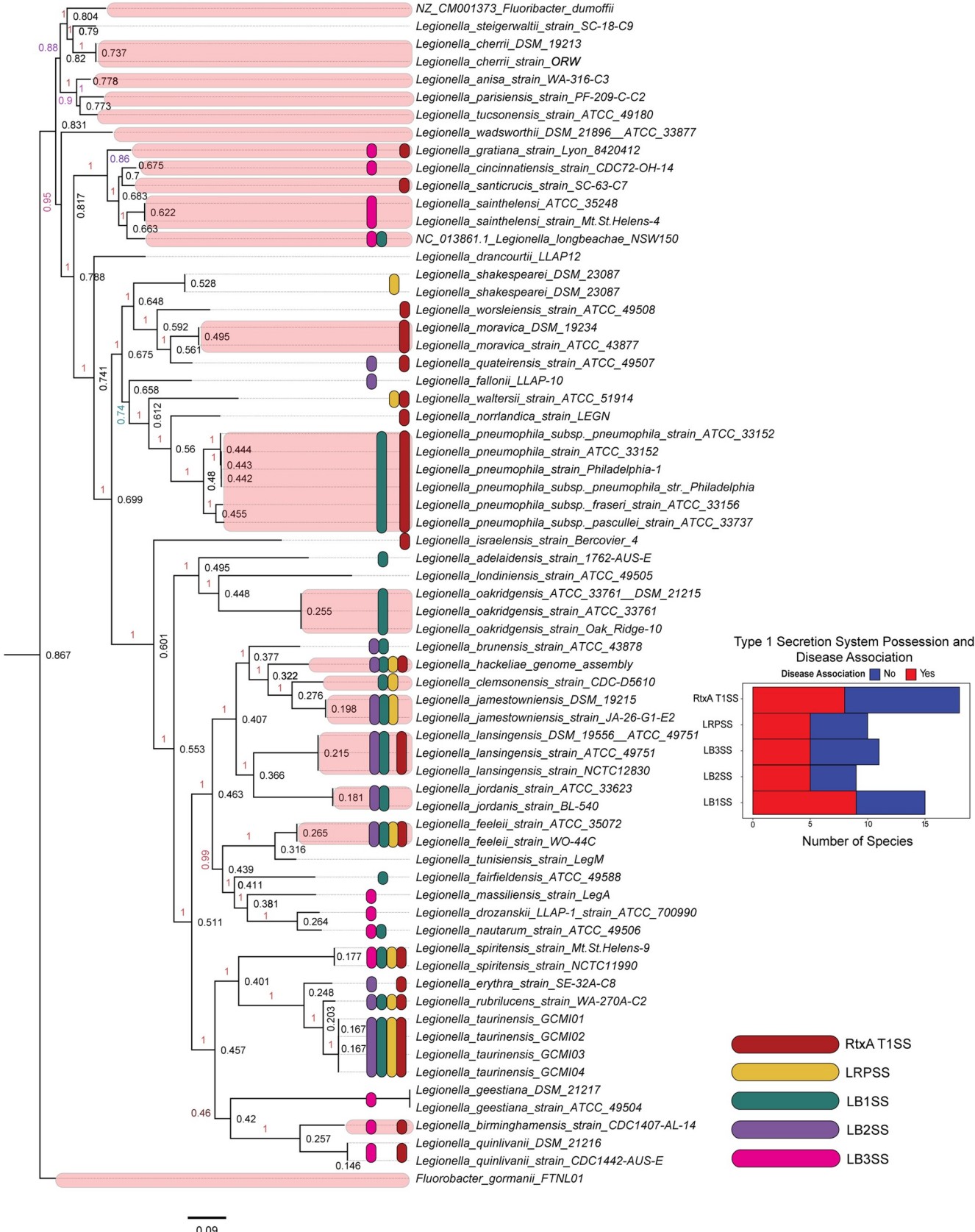

**Fig 4. Nucleotide whole genome sequence FastTree tree predicted using a core set of marker genes predicted by PhyloSift of *Legionella* spp. overlaid with the distribution of T1SSs.** Pink color indicates species that have been associated with human infection [27].

caused by *L. pneumophila*, 3 (1%) by *L. longbeachae*, 4 (1.3%) by *L. micdadei*, 1 (0.3%) by *L. bozemanii* and 3 (1%) by *L. feeleii*. One hundred and eight (35%) were due to unreported species of *Legionella*. Of these, only *L. feeleii* and possibly some strains of *L. longbeachae* possess the RtxA T1SS. Additionally, *Gomez-Valero* et al 2019 measured replicative capacity of different *Legionella* spp. in THP-1 macrophages and found that *L. jordanis*, *L. taurinensis*, *L. jamestowniensis*, *L parisiensis*, *L brunensis*, and *L. bozemanii* displayed replicative capacity similar or superior to that of *L. pneumophila* [29]. Of these species, *L. taurinensis*, *L. brunensis*, and *L. jamestowniensis*, (three of six (50%)) encode the RtxA T1SS. Therefore, while many *Legionella* spp. possess the T1SS responsible for RtxA translocation, our results indicate that the system alone does not predict propensity for disease association or intracellular replication in macrophages.

In conclusion, we report that the RtxA T1SS and four novel T1SSs discovered in *L. taurinensis* are broadly distributed among *Legionella* species and provide the first extensive survey and classification of T1SSs in the *Legionella* genus. Sequence and phylogenetic analysis indicate that *Legionella* spp. encode T1SSs that facilitate diverse functions, including bacteriocin transport and protein secretion. Importantly, no relationship was detected between the possession of the RtxA T1SS with disease association, drawing into question the nature of RtxA and its T1SS as a genetic virulence determinant.

## Methods

### Determining epidemiological features of *Legionella* spp.

We considered a *Legionella* species to be "disease-associated" based on whether not any strain of the species has ever been isolated from a patient. To determine this, we referenced the online resource at https://www.specialpathogenslab.com/legionella-species.php (accessed late 2018—mid 2019) [27] and performed a literature review on PubMed Central to confirm accuracy.

### Sampling and culture methods

Two one-liter samples of water were collected from one tap of a school located in Genesee County serviced by a groundwater well in March of 2016 into sterile polypropylene bottles (Nalgene, Rochester, NY) with 24 mg of sodium thiosulfate per liter added to quench chlorine for preservation prior to microbial analysis. Samples were collected with the permission of the community school board supervisor. Within 24 hours, samples were filter-concentrated onto a sterile 0.22 μm pore size mixed-cellulose ester membrane (Millipore, Billerica, MA) and resuspended in 5 mL sterile tap water prior to culturing according to International Standards of Organization (ISO) methods [30] for the recovery of *L. pneumophila* on highly selective buffered charcoal yeast extract media with supplemented glycine, polymyxin B sulfate, cycloheximide and vancomycin.

### Whole genome sequencing, assembly, and annotation

DNA was extracted using FastDNA SPIN Kit (MP Biomedicals, Solon, OH) according to manufacturer instructions. Purified DNA was quantified via a Qubit 2.0 Fluorometer (Thermo Fisher, Waltham, MA) and analyzed via gel electrophoresis to verify DNA integrity. Sequencing was conducted by MicrobesNG (Birmingham, United Kingdom) on a MiSeq platform

(Illumina, San Diego, CA) with 2 x 250 bp paired-end reads. Libraries were constructed using a modified Nextera DNA library preparation kit (Illumina, San Diego, CA). Reads were trimmed using Trimmomatic [31] and *de novo* assemblies were generated using SPAdes [32]. This Whole Genome Shotgun project has been deposited at DDBJ/ENA/GenBank under the accession PRJNA453403. The version described in this paper is version PRJNA453403.

## Sequence and phylogenetic analysis

The initial bioinformatic analysis that detected the T1SSs was performed using the integrated microbial genomes database and comparative analysis system (IMG) from the Joint Genome Institute [33].

To detect the RtxA T1SS components, *L. pneumophila* LssB and LssD amino acid sequences were compared with the protein sequences of *Legionella* spp. using blastp. For the four novel T1SSs, *L. taurinensis* amino acid sequences were used as query. A BLAST result was used to support a gene name when the amino acid sequence had overall identity of ≥40% (adjusted for incomplete query cover) with one of the query sequences. This criterion was used for all genes, except *lb1ssD*, for which many species displayed <40% amino acid homology relative to *L. taurinensis lb1ssD* (S1 File Table B). Despite this, these genes were consistently found to be co-localized with *lb1ssB* homologs with ≥40% homology to *L. taurinensis* (for instance, WP_012979428.1/WP_012979429.1 in *L. longbeachae* strain NSW150). The 40% identity cut-off is used by the Enzyme Commission to establish functional conservation between similar proteins [34,35]. Further, Qin *et al.* 2017 used the 40% identity cut-off previously to investigate the distribution of secretion systems in *Legionella* species [24]. Last, protein phylogeny of the ABC transporters was inferred to validate the results suggested by the blastp results. This analysis resulted in the renaming of several T1SSs which were near 40% homologous (S1 File Table B). *Legionella* T1SS sequences with suggested names based on the results of the protein phylogeny and the sequence identity analysis are compiled (S1 File Table B).

For the maximum likelihood (ML) tree, 187 amino acid sequences of T1SS ABC transporters were used. The sequences of previously described non-*Legionella* T1SS ABC transporters were chosen based on a review of the literature, especially [3,4,26]. These sequences were then compared with the non-redundant protein database [36] using blastp and the ten best hits for each non-*Legionella* T1SS ABC transporter were collected. These sequences and the *Legionella* sequences were then aligned using MUSCLE v3.8.31 [37] with default settings. The aligned sequences were then trimmed using the heuristic method for trimAl, which resulted in a 482 amino acid aligned region (available in online materials) [38]. Maximum likelihood trees for the trimmed sequence alignments were created using the RAxML webserver [39] with default settings including the GAMMA model of rate heterogeneity, the LG amino acid substitution matrix, and automatic bootstrapping (bootstopping cut-off = 0.03). The most likely tree was overlaid with bipartition values of 200 bootstrap replicates at the command line using RAxMLHPC v8.2.4. The tree constructed from the trimmed sequences is displayed in Fig 3.

## Whole genome sequence tree

Reference genomes from 45 *Legionella* species were downloaded from NCBI (full list and sequences available at https://doi.org/10.6084/m9.figshare.9162152.v1) A set of core marker genes were identified using PhyloSift [40] and hmmer [41] to build a multiple sequence alignment. This alignment was used by FastTree [42] to generate a phylogenetic tree.

## Availability of data and material

All data including all *Legionella* sequences identified in the present study are provided in the paper and its supplementary materials, and may be accessed online through FigShare at doi:10.6084/m9.figshare.9162161.v1, doi:10.6084/m9.figshare.9162146.v1, doi:10.6084/m9.figshare.9162152.v1, doi:10.6084/m9.figshare.9162161.v1. The Whole Genome Shotgun project has been deposited at DDBJ/ENA/GenBank under the accession PRJNA450138. The version described in this paper is version PRJNA450138.

## Supporting information

**S1 File. Supplementary information.** Supplementary figures and tables. PONE-D-19-25375R1_Supplementary_Information.
(DOCX)

## Acknowledgments

The authors acknowledge and appreciate the time and support provided by Drs. William Rhoads and Marc Edwards.

## Author Contributions

**Conceptualization:** Connor L. Brown, Emily Garner, Guillaume Jospin, David A. Coil, Amy J. Pruden.

**Data curation:** Emily Garner.

**Formal analysis:** Connor L. Brown, Guillaume Jospin, Biswarup Mukhopadhyay.

**Investigation:** Guillaume Jospin, David O. Schwake.

**Methodology:** Emily Garner, Guillaume Jospin, David A. Coil, David O. Schwake, Jonathan A. Eisen, Biswarup Mukhopadhyay.

**Project administration:** David A. Coil.

**Resources:** David O. Schwake.

**Supervision:** Jonathan A. Eisen, Amy J. Pruden.

**Visualization:** Connor L. Brown, Guillaume Jospin.

**Writing – original draft:** Connor L. Brown, Biswarup Mukhopadhyay.

**Writing – review & editing:** Connor L. Brown, Emily Garner, Guillaume Jospin, David A. Coil, David O. Schwake, Jonathan A. Eisen, Biswarup Mukhopadhyay, Amy J. Pruden.

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
