## [Decision Letter · Decision Letter 0]

14 Oct 2019

PONE-D-19-25375

Whole genome sequence analysis reveals broad distribution of the RtxA type 1 secretion system and four novel type 1 secretion systems throughout the Legionella genus

PLOS ONE

Dear Dr. Pruden,

Thank you for submitting your manuscript to PLOS ONE. After careful consideration, we feel that it has merit but does not fully meet PLOS ONE’s publication criteria as it currently stands. Therefore, we invite you to submit a revised version of the manuscript that addresses the points raised during the review process.

We would appreciate receiving your revised manuscript by Nov 28 2019 11:59PM. To enhance the reproducibility of your results, we recommend that if applicable you deposit your laboratory protocols in protocols.io, where a protocol can be assigned its own identifier (DOI) such that it can be cited independently in the future. For instructions see: http://journals.plos.org/plosone/s/submission-guidelines#loc-laboratory-protocols

We look forward to receiving your revised manuscript.

Kind regards,

Daniel E. Voth, Ph.D.

Academic Editor

PLOS ONE

Journal Requirements:

Additional Editor Comments (if provided):

Reviewers' comments:

Reviewer's Responses to Questions

**Comments to the Author**

1. Is the manuscript technically sound, and do the data support the conclusions?

Reviewer #1: Yes

Reviewer #2: Partly

2. Has the statistical analysis been performed appropriately and rigorously? 

Reviewer #1: Yes

Reviewer #2: Yes

3. Have the authors made all data underlying the findings in their manuscript fully available?

Reviewer #1: Yes

Reviewer #2: Yes

4. Is the manuscript presented in an intelligible fashion and written in standard English?

Reviewer #1: Yes

Reviewer #2: Yes

5. Review Comments to the Author

Reviewer #1: This is a well organized report on the prevalence of T1SS in isolates of Legionella. Genome sequence was generated from specimen isolated in Flint, Michigan groundwater. Four isolates were described with one novel genome revealed.

The emphasis of the study examines distribution of RtxA-like type 1 secretion systems, with the important conclusion that this system is not limited to distribution among human pathogenic strains of Legionella as previously proposed.

The majority of the paper reads like a comprehensive review of classes of T1SS, focusing on classifying new secretion systems in L.taurinensis.

The use of 'virulence factor' throughout to describe the RtxA locus in L.pneumophila is somewhat generous. When one examines the papers that this study cites, nearly all phenotypes said to be associated with the rtxA locus present as less than two-fold differences in most experiments. of the three enhanced entry loci described in reference 15, the rtxA-containing locus has not been pursued since these publications. Other enhanced entry (enh) have been explored.

The wording in the abstract (lines 41-42) 'conserved virulence factor in L.pneumophila' doesn't convey the message correctly. It should be changed to 'virulence factor exclusive to L.pneumophila'.

Reviewer #2: This manuscript describes an in silico analysis of predicted type I secretion system (T1SS) distribution in multiple Legionella species. Brown et al. describe isolation of four new strains of Legionella taurinensis from Genesee county Michigan. Upon genome sequencing of these strains, the authors identified four putative T1SSs, one with homology to the RtxA T1SS and three putative novel T1SSs. They also probed the genomes of 45 additional Legionella species for the presence of T1SSs using amino acid sequence homology as an indicator. They found predicted RtxA and novel LB2SS T1SS in previously sequenced Legionella species and determined that distribution of T1SS is not a predictor of virulence. This is an interesting study that re-evaluates the distribution of T1SS within Legionella.

Comments:

The title should be changed to ..”four putative novel type 1 secretion…”. There is no experimental evidence demonstrating that these are functional type 1 secretion systems.

Table 1 is missing from the submission files.

As there is a virulence defect observed for loss of the rtxA T1SS in L. pneumophila (Cirillo et al, 2001), the authors should rephrase the sentence in lines 56-57. The RtxA system could be a virulence determinant, it’s just not sufficient if other virulence factors are not present/active, etc. Another example is that the Dot/Icm secretion system is present in all sequences strains of Legionella but they're not all pathogenic to humans.

Line 96: add the word “potential” or “putative” in front of “novel T1SS”

Line 103: Remove “(Garner et al. in review)” unless the study has been published.

Figure 2: B and C are switched in the figure legend and text.

Out of curiosity, any idea what the hypothetical genes in the L. taurinensis lssXYZABD locus could be doing?

Line 130: How did you decide on a 40% identity cut-off? In your predictions, was the rtxA substrate homolog also identified?

Line 147: The blastp algorithm was used here, correct? If so, please specify.

Line 200: Please reference the sentence starting “Of these 19,…”.

Line 216: Please reference the statement ending in “…Legionella tested (n = 10)”.

Line 229: please change the language in the sentence starting with “In contrast,…” to emphasize that presence if these systems is predicted based on homology.

Figure 4 is of poor quality (pixelated), please improve the quality of this image to increase its readability. Also, what does the light pink color indicate?

Please also keep in mind that just because a Legionella species has not been isolated from a patient, it does not have potential to cause disease. It is likely that some species may be less able to colonize sources of human infection.

6. PLOS authors have the option to publish the peer review history of their article (what does this mean?). If published, this will include your full peer review and any attached files.

Reviewer #1: No

Reviewer #2: No

---

## [Author Response · Author response to Decision Letter 0]

10 Dec 2019

PONE-D-19-25375

Whole genome sequence analysis reveals broad distribution of the RtxA type 1 secretion system and four novel type 1 secretion systems throughout the Legionella genus

PLOS ONE

Dear Dr. Pruden,

Thank you for submitting your manuscript to PLOS ONE. After careful consideration, we feel that it has merit but does not fully meet PLOS ONE’s publication criteria as it currently stands. Therefore, we invite you to submit a revised version of the manuscript that addresses the points raised during the review process.

We would appreciate receiving your revised manuscript by Nov 28 2019 11:59PM. To enhance the reproducibility of your results, we recommend that if applicable you deposit your laboratory protocols in protocols.io, where a protocol can be assigned its own identifier (DOI) such that it can be cited independently in the future. For instructions see: http://journals.plos.org/plosone/s/submission-guidelines#loc-laboratory-protocols

• A rebuttal letter that responds to each point raised by the academic editor and reviewer(s). This letter should be uploaded as separate file and labeled 'Response to Reviewers'.

• A marked-up copy of your manuscript that highlights changes made to the original version. This file should be uploaded as separate file and labeled 'Revised Manuscript with Track Changes'.

• An unmarked version of your revised paper without tracked changes. This file should be uploaded as separate file and labeled 'Manuscript'.

We look forward to receiving your revised manuscript.

Kind regards,

Daniel E. Voth, Ph.D.

Academic Editor

PLOS ONE

[Note: Line references refer to the revised version of the manuscript, rather than the version showing tracked changes.]

Reviewers’ Comments to Authors: 

Reviewer #1: This is a well-organized report on the prevalence of T1SS in isolates of Legionella. Genome sequence was generated from specimen isolated in Flint, Michigan groundwater. Four isolates were described with one novel genome revealed. The emphasis of the study examines distribution of RtxA-like type 1 secretion systems, with the important conclusion that this system is not limited to distribution among human pathogenic strains of Legionella as previously proposed. The majority of the paper reads like a comprehensive review of classes of T1SS, focusing on classifying new secretion systems in L.taurinensis.

1. The use of 'virulence factor' throughout to describe the RtxA locus in L. pneumophila is somewhat generous. When one examines the papers that this study cites, nearly all phenotypes said to be associated with the rtxA locus present as less than two-fold differences in most experiments. of the three enhanced entry loci described in reference 15, the rtxA-containing locus has not been pursued since these publications. Other enhanced entry (enh) have been explored.

We agree that the nature of RtxA as a virulence factor is uncertain. However, several experimental studies since Cirillo et al. 2001 have supported the role of RtxA and its type 1 secretion system (T1SS) as a possible virulence factor (Cirillo et al. 2002, Fuche et al 2015, Qin et al. 2017) and these observations have been reported in a substantial review (Smith et al. 2019). Of note, Cirillo et al. 2002 examined 20 non-pneumophila Legionella genomes and found that RtxA was restricted to L. pneumophila and L. feeleii; this led the authors to conclude:

“This observation, along with previous studies demonstrating that this gene plays an important role in virulence (Cirillo et al., 2001), has helped to fulfill ‘molecular Koch’s postulates’ for rtxA, supporting the conclusion that this gene plays a role in disease.”

Therefore, we argue that the use of “virulence factor” is justified to reflect the assumptions of the current available literature. However, as these studies were not conducted in animal models, we have updated the first paragraph of the discussion (lines 234-236) to reflect the spirit of the reviewer’s comment: 

Thus, the emergent consensus model has been that the RtxA system is an important conserved genetic virulence determinant unique to L. pneumophila, despite the absence of additional experimental investigation of its function in animal models since the discovery of the enhanced entry locus in 2001 [15].

2. The wording in the abstract (lines 41-42) 'conserved virulence factor in L.pneumophila' doesn't convey the message correctly. It should be changed to 'virulence factor exclusive to L. pneumophila'.

The wording has been changed to reflect that it is a virulence factor exclusive to L. pneumophila rather than being conserved (lines 40-41). 

Reviewer #2: This manuscript describes an in-silico analysis of predicted type I secretion system (T1SS) distribution in multiple Legionella species. Brown et al. describe isolation of four new strains of Legionella taurinensis from Genesee county Michigan. Upon genome sequencing of these strains, the authors identified four putative T1SSs, one with homology to the RtxA T1SS and three putative novel T1SSs. They also probed the genomes of 45 additional Legionella species for the presence of T1SSs using amino acid sequence homology as an indicator. They found predicted RtxA and novel LB2SS T1SS in previously sequenced Legionella species and determined that distribution of T1SS is not a predictor of virulence. This is an interesting study that re-evaluates the distribution of T1SS within Legionella.

1. The title should be changed to ”four putative novel type 1 secretion…”. There is no experimental evidence demonstrating that these are functional type 1 secretion systems.

The title has been updated to reflect that these systems are putative. 

2. Table 1 is missing from the submission files.

This table has been uploaded along with the resubmission. 

3. As there is a virulence defect observed for loss of the rtxA T1SS in L. pneumophila (Cirillo et al, 2001), the authors should rephrase the sentence in lines 56-57. The RtxA system could be a virulence determinant, it’s just not sufficient if other virulence factors are not present/active, etc. Another example is that the Dot/Icm secretion system is present in all sequences strains of Legionella but they're not all pathogenic to humans. 

We have updated the final line of the abstract (lines 56-57): 

“These results draw into question the nature of RtxA and its T1SS as a singular genetic virulence determinant. Future investigations should focus on mechanistic explanations for the association of RtxA with virulence.”

4. Line 96: add the word “potential” or “putative” in front of “novel T1SS”

This line has been updated (line 96). 

5. Line 103: Remove “(Garner et al. in review)” unless the study has been published.

This reference has been updated (lines 103). 

6. Figure 2: B and C are switched in the figure legend and text.

This has been fixed (lines 111-117). 

7. Out of curiosity, any idea what the hypothetical genes in the L. taurinensis lssXYZABD locus could be doing?

These proteins appear to be part of a zeta toxin/antitoxin system. The only homologs of these genes we could find were in Legionella species, and the colocalization of the two genes was preserved in all Legionella genomes we looked at. Analyzing the amino acid sequence of the larger protein, we found evidence of an NTP-ase p-loop domain associated with UDP-kinase/zeta toxin activity (domain IPR010488). On the Integrated Microbial Genomes repository, many of the homologs of the larger protein were annotated as putative zeta toxins. 

8. Line 130: How did you decide on a 40% identity cut-off? In your predictions, was the rtxA substrate homolog also identified?

The 40% identity cut-off is used by the Enzyme Commission and, when comparing domains with the same fold, reflects specific functional conservation well (Wilson et al. 2000). Further, Qin et al. 2017 used this cut-off value to identify homologs of secretion system components in Legionella spp. 

This text (and appropriate citations) was added to reflect this (lines 298-301): 

“The 40% identity cut-off is used by the Enzyme Commission to establish functional conservation between similar proteins (33, 34) Further, Qin et al. 2017 used the 40% identity cut-off previously to investigate the distribution of secretion systems in Legionella species (23).” 

9. Line 147: The blastp algorithm was used here, correct? If so, please specify.

This line was updated with the search algorithm used (tblastn) (lines 146-147). 

10. Line 200: Please reference the sentence starting “Of these 19,…”.

A reference was added to this line (lines 202). 

11. Line 216: Please reference the statement ending in “…Legionella tested (n = 10)”.

A reference was added to this line (line 222). 

12. Line 229: please change the language in the sentence starting with “In contrast,…” to emphasize that presence if these systems is predicted based on homology.

This sentence was changed to (lines 236-238):

“In contrast, we found bioinformatic evidence that homologs of the RtxA T1SS and the four novel T1SSs are prevalent throughout the genus, even among species not presently known to cause disease.”

13. Figure 4 is of poor quality (pixelated), please improve the quality of this image to increase its readability. Also, what does the light pink color indicate?

The resolution of Figure 4 has been improved and a sentence was added to the figure legend of Figure 4 (line 197-198): 

“Pink color indicates species that have been associated with human infections (27)” 

Please note: the version of the figure within the PLOS submission PDF has less resolution than the actual TIF file. To see the actual figure, please use the download link in the top right corner of the submission document. 

14. Please also keep in mind that just because a Legionella species has not been isolated from a patient, it does not have potential to cause disease. It is likely that some species may be less able to colonize sources of human infection.

We agree with the reviewer’s comment and have introduced the following changes to the end of the results section (lines 212-217): 

“However, our observation brings up the possibility that some of the Legionella species that thus far have not been isolated from patients are inefficient colonizers of built environments that are common exposure routes and, therefore, remain classified as non-pathogenic.”

---

## [Editor Report · Decision Letter 1]

18 Dec 2019

Whole genome sequence analysis reveals the broad distribution of the RtxA type 1 secretion system and four novel putative type 1 secretion systems throughout the Legionella genus

PONE-D-19-25375R1

Dear Dr. Pruden,

We are pleased to inform you that your manuscript has been judged scientifically suitable for publication and will be formally accepted for publication once it complies with all outstanding technical requirements.

With kind regards,

Daniel E. Voth, Ph.D.

Academic Editor

PLOS ONE
---

## [Editor Report · Acceptance letter]

27 Dec 2019

PONE-D-19-25375R1 

Whole genome sequence analysis reveals the broad distribution of the RtxA type 1 secretion system and four novel putative type 1 secretion systems throughout the *Legionella* genus 

Dear Dr. Pruden:

I am pleased to inform you that your manuscript has been deemed suitable for publication in PLOS ONE. Congratulations! Your manuscript is now with our production department. 

With kind regards,

on behalf of

Dr. Daniel E. Voth 

Academic Editor

PLOS ONE